# Automated and Continuous Fatigue Monitoring in Construction Workers Using Forearm EMG and IMU Wearable Sensors and Recurrent Neural Network

**DOI:** 10.3390/s22249729

**Published:** 2022-12-12

**Authors:** Srikanth Sagar Bangaru, Chao Wang, Fereydoun Aghazadeh

**Affiliations:** 1ISC Constructors LLC, 20480 Highland Rd, Baton Rouge, LA 70817, USA; 2Bert S. Turner Department of Construction Management, Louisiana State University, 3315D Patrick F. Taylor Hall, Baton Rouge, LA 70803, USA; 3Department of Mechanical & Industrial Engineering, Louisiana State University, 3250A Patrick F. Taylor Hall, Baton Rouge, LA 70803, USA

**Keywords:** fatigue monitoring, construction labor shortage, muscle activity, activity recognition, oxygen prediction, wearable sensor, aerobic fatigue threshold, scaffold building, work-related musculoskeletal disorders

## Abstract

About 40% of the US construction workforce experiences high-level fatigue, which leads to poor judgment, increased risk of injuries, a decrease in productivity, and a lower quality of work. Therefore, it is essential to monitor fatigue to reduce its adverse effects and prevent long-term health problems. However, since fatigue demonstrates itself in several complex processes, there is no single standard measurement method for fatigue detection. This study aims to develop a system for continuous workers’ fatigue monitoring by predicting the aerobic fatigue threshold (AFT) using forearm muscle activity and motion data. The proposed system consists of five modules: Data acquisition, activity recognition, oxygen uptake prediction, maximum aerobic capacity (MAC) estimation, and continuous AFT monitoring. The proposed system was evaluated on the participants performing fourteen scaffold-building activities. The results show that the AFT features have achieved a higher accuracy of 92.31% in assessing the workers’ fatigue level compared to heart rate (51.28%) and percentage heart rate reserve (50.43%) features. Moreover, the overall performance of the proposed system on unseen data using average two-min AFT features was 76.74%. The study validates the feasibility of using forearm muscle activity and motion data to workers’ fatigue levels continuously.

## 1. Introduction

The construction industry is one of the leading industries in the world, which spends $10 trillion on construction-related goods and services annually [1]. However, the construction industry faces a massive workforce shortage of skilled craft workers [2]. More than eight out of ten construction firms report having a hard time finding qualified workers. One of the significant causes of workforce shortage is the premature retirement of skilled craft workers due to safety and health issues. Due to a lack of proper safety training and monitoring systems, the construction workforce is exposed to fatal and non-fatal injuries such as Work-related Musculoskeletal Disorders (WMSDs). According to the report released by the International Labor Organization (ILO) in 2015, it was estimated that there are at least 60,000 construction-related fatalities all over the world each year [3]. Similarly, in the United States, 971 (18.9%) out of 5147 fatal injuries occurred in the construction industry in 2017, based on occupational injury reports released by the Bureau of Labor Statistics [4]. In addition, the estimated non-fatal injury rate in the construction industry was 5.3 cases per 10,000 full-time workers in 2018 [5]. According to the 2018 Liberty Mutual Workplace Safety Index, businesses spent more than one billion dollars per week for non-fatal severe workplace injuries. The high rate of non-fatal injuries in the construction industry is mainly due to WMSDs. WMSDs are among construction workers’ most prevalent occupational health problems due to highly labor-intensive construction tasks.

The construction industry often involves highly labor-intensive and repetitive tasks, which results in worker physical fatigue. About 40% of the US construction workforce experiences high-level fatigue, leading to poor judgment, increased risk of injuries, decreased productivity, and a lower quality of work [6,7]. Further, excessive fatigue due to working in unpleasant conditions, long working hours, and heavy workloads can aggravate fatigue’s adverse effects and lead to WMSDs and productivity loss. Moreover, fatigue has been shown to result in impairing physical and cognitive functions [8] and identified as a possible risk factor for slip-induced falls, which is one of the “fatal four” causes of fatalities in the construction industry, according to Occupational Safety and Health Administration [9].

A significant number of craft workers (20% to 40%) routinely exceed generally accepted physiological thresholds for manual work shifts [6]. Physical fatigue and impaired mental capacity lead to a high risk of accidents in any environmental condition [10] and affects workers’ safety performance [11]. Since physical fatigue is a predominant risk factor for injuries and illnesses in the construction industry, it is essential to monitor fatigue to reduce its adverse effects and prevent long-term health problems. However, since fatigue demonstrates itself in several complex processes, there is no single standard measurement method for fatigue detection. For example, if a specific physiological function is altered, it only reflects the body’s adaptive behavior instead of the level of fatigue [12,13].

Moreover, overall physical fatigue is a result of the interaction between local (muscular fatigue) and central factors (such as metabolic, cardiovascular, and thermoregulatory) [14]. Therefore, fatigue quantification typically involves a combination of kinematic and kinetic measurements, often supplemented or substituted by physiological (body temperature, heart rate, or muscle activity) and subjective measures (perceived exertion or discomfort). Physical fatigue is always associated with a high workload as physical demand. The evaluation of measured workloads involves two phases: assessment and evaluation. The assessment phases involve measuring physiological response to work as a measure of physical demand. The evaluation phase involves determining whether a task’s physical demand (workload) is excessive and workers performing the task may suffer from physical fatigue. The physical demand evaluation techniques include the classification of work severity based on recommendations for oxygen uptake, energy expenditure, and heart rate and the evaluation of physical fatigue based on aerobic fatigue threshold, absolute energy expenditure, and heart rate limits [6].

Even though there is no gold standard for fatigue measurement, several subjective and objective techniques are adapted for occupational use. The subjective evaluation of fatigue involves workers’ feedback to the questionnaire, and several construction studies used various fixed sets of questions and feedback scales related to fatigue [11,15,16,17,18,19,20,21]. However, subjective assessments rely on workers’ internal perceptions, previous experiences and interrupt the ongoing work. Furthermore, most of these measurement techniques are cumbersome and impractical on construction sites, emphasizing the need for a continuous fatigue monitoring system with minimal obstruction to construction tasks [10,22].

With advancements in wearable sensing technology, a few researchers have developed objective techniques using physiological sensors to assess the workers’ overall physical fatigue by monitoring the physiological responses of the worker to physical demand [10,13,22,23]. In recent studies, Jebelli, Choi and Lee [22] have recognized physical demand during on-site work by training the machine learning model on workers’ photoplethysmogram (PPG), electrodermal activity (EDA), and skin temperature (ST) with an energy expenditure of the task, which was determined using the Energy Expenditure Prediction Program (EEPP). Aryal, Ghahramani and Becerik-Gerber [10] used skin temperature and heart rate for fatigue detection based on workers’ ratings of perceived exertion. Hwang and Lee [13] used heart rate reserve (%HRR) as a metric to distinguish different levels of physical demand. Maman, et al. [24] have estimated the RPE fatigue level of an individual performing assembly, manual material handling, and supply pick-up tasks using four inertial measurement units (IMU) attached to the human body. Even though these studies have established the potential of physiological responses to determine the worker’s physical demand for a long duration, they still have limitations, such as being unable to identify the physical demand of individuals with different characteristics (such as work experience, work conditions, age, and health status), previous studies are limited to classifying individual physical demand based on work severity, not capable of continuous workers’ fatigue level monitoring for multiple tasks performed in short intervals, the measurements such as heart rate, skin temperature, and electrodermal activity are highly influenced by external factors, which may not yield reliable results on construction sites, and an inability to determine direct impacts of fatigue on construction activities. Moreover, the previous studies focused on work severity classification rather than determining the workers’ real-time physical fatigue.

To overcome these challenges or limitations, this study proposes an automated continuous workers’ fatigue monitoring system by measuring aerobic fatigue threshold (AFT) using forearm muscle activity and kinematic data for an activity. The aerobic fatigue threshold has been for the proposed system because AFT is activity-dependent, which is appropriate for construction. Unlike heart rate, electrodermal activity, and skin temperature, which are highly influenced by external factors, the forearm muscle activity, and kinematic data are activity specific. Since the proposed system is workers’ activity-centric, it is highly suitable for construction workers’ fatigue monitoring as they are involved in various labor-intensive tasks throughout the day.

## 2. Literature Review

### 2.1. Definition and Causes of Fatigue

Fatigue is a declination of a person’s ability to maintain a normal level of performance and impaired mental alertness [25]. In general, fatigue is defined as a state of feeling tired, sleepy, or weary and results from loss of sleep, an extended period of anxiety, exposure to an adverse environment, and prolonged physical and mental work. The unidimensional fatigue characterization usually describes it as mental and physical fatigue [26]. Mental fatigue results in a decrease in cognitive and behavioral performance, whereas physical fatigue leads to a decline in the capacity to perform physical activity [27]. Fatigue is a complex phenomenon caused by various factors in the workplace and outside of the workplace. Outside the workplace, the lack of restorative sleep is the most common cause of fatigue. Whereas in the workplace, fatigue is caused due to excess physical and mental workload. The workload refers to the work assigned to a worker, categorized into a physical load, a metal load, and an environmental load [28]. The fatigue might be due to an individual factor or a combination of interrelated factors. Figure 1 shows the work-related causes of fatigue [29]. Moreover, physical fatigue is identified as localized muscular fatigue and overall physical fatigue. Compared to localized muscular fatigue, overall physical fatigue is challenging to quantify as it is caused by the interactions between local (muscular) and central (metabolic, thermoregulatory, cardiovascular, etc.) factors.

### 2.2. Current Approaches for Fatigue Measurement

Since the human body demonstrates physical fatigue in several ways, there exist numerous ways to measure fatigue [30]. However, these methods are limited in application since they are developed for specific contexts and purposes [31]. The fatigue measurement methods adopted for occupational use can be broadly classified into subjective and objective assessment techniques (Figure 2).

Early attempts at measuring fatigue involve subjective assessment using a fixed questionnaire related to physical and mental fatigue [32,33]. Several studies in construction used different questionnaires and subjective feedback scales to quantify fatigue involved in construction activities [11,20,21,31,34]. Fang, Jiang, Zhang and Wang [11] have developed an experimental method to understand the effect of fatigue on construction workers’ safety performance where the authors used the Fatigue Assessment Scale for Construction Workers (FASCW) developed by Zhang, Sparer, Murphy, Dennerlein, Fang, Katz and Caban-Martinez [31] to determine the fatigue level. The experimental study has concluded that above fatigue level 20, there was a linear relationship between workers’ fatigue and error rate (a measure of safety performance). Mitropoulos and Memarian [20] used the NASA Task load index (TLX) rating scale to determine the task demands in masonry work, where NASA TLX measures mental load, physical load, temporal load, and performance of the worker in a particular task. Measuring the TLX index facilitated the determination of various factors, such as task features, supervisor practices, and work conditions, which yield high task demands [20]. Chan, Yi, Wong, Yam and Chan [21] used the Physiological Strain Index (PSI) to determine the recovery time after the fatigue state, which was identified by the ratings of perceived exertion (RPE) using the Borg CR10 Scale. Yi, Chan, Wang and Wang [34] developed an early-warning system to monitor workers’ heat-strain levels when working in a hot and humid environment using subjective index perception rating of perceived exertion (RPE) and artificial neural network (ANN). The ANN-based prediction model in the early-warning system uses wet bulb globe temperature (WBGT), age, BMI, job nature, work duration, alcohol drinking habit, and smoking habit as input features to predict RPE to monitor workers’ heat-strain level. However, subjective fatigue assessment has two significant limitations for the field. First, the feedback assessment is strongly biased due to the workers’ internal perceptions, ethics, and socioeconomic backgrounds [31]. Second, the subjective feedback collection on construction sites by stopping the worker while performing a task is cumbersome and not practical.

The objective measurement of overall physical fatigue involves quantifying workers’ physiological processes and kinematic data. The physiological processes involve heart rate, energy consumption, oxygen consumption, and EMG activity, whereas the kinematic data includes the body motion data collected using motion capture systems such as kinetic camera and inertial measurement unit sensors (IMUs). Optoelectrical measurement systems are considered the gold standard for body motion analysis within a research setting. However, due to the high cost, large installation spaces, and extensive post-processing of optoelectrical measurement systems, IMU sensors are widely used for full-body motion data collection. The IMUs are non-intrusive wearable sensors integrated with accelerometers, gyroscopes, and magnetometers to measure the body segments’ acceleration, orientation, and velocity. IMUs are used for overall physical fatigue detection by monitoring the reduction of motor control [35]. Motor performance and control are assessed using the motion smoothness metrics such as the ratio between the maximum and the mean velocity during the movement, the number of peaks in the velocity profile, and jerk derived from kinematic data [36,37]. Jerk is the first derivative of acceleration used to determine motor control and motion smoothness. Van Dieën, et al. [38] investigated that jerk at various joints, such as the ankle, hip, knee, and lumbosacral joint, is increased due to fatigue during the repetitive lifting of a barbell. Maman, Yazdi, Cavuoto and Megahed [24] developed logistic and MLR-based physical fatigue detection models using features that included wrist and hip jerks during simulated manufacturing tasks. The features from the sensor data are extracted using the Least Absolute Shrinkage and Selection Operator (LASSO). The study reported that the accelerometer located at the hip and wrist are strong predictors of physical fatigue than heart rate features. Zhang, Diraneyya, Ryu, Haas and Abdel-Rahman [35] investigated the feasibility of using jerk as the metric to detect physical fatigue in repetitive bricklaying activity. The results indicate that the jerk values obtained from the upper arms and pelvis are significant compared to the values from the hands and forearms. However, the motion smoothness metrics, such as the jerk values, are task-dependent and highly influenced by the worker’s repeated shocks, impacts, and skill level.

Harnessing the workers’ physiological processes, such as oxygen consumption, heart rate, skin temperature, muscle engagement, and blood pressure, determines physical workload or fatigue level [6,23,39,40,41]. Measuring the physiological workload can assess the level of physical fatigue. The physiological workload can be determined by measuring oxygen uptake while performing work. With advancements in wearable sensing technologies and machine learning, the oxygen uptake or VO_2_ can be estimated using sensor data such as heart rate and IMU. According to NIOSH recommendation, the average oxygen uptake during an eight hour workday should not exceed 33% of activity-specific maximum aerobic capacity [42,43,44]. Abdelhamid and Everett [6] reported that 20–40% of craft workers exceed physiological thresholds daily by measuring the workers’ oxygen uptake and heart rate. 

Most construction studies focused on measuring workload and work severity classification based on physiological responses [6,23,45,46]. Abdelhamid and Everett [6] used oxygen uptake and heart rate to determine the physical demands required for different construction activities. Wong del, Chung, Chan, Wong and Yi [45] proved that the energy required to perform bar fixing tasks was more than bar bending tasks in a hot and humid environment. Chan, Yi, Wong, Yam and Chan [21] determined the optimal recovery time for rebar workers after working to exhaustion in a hot and humid environment using blood pressure, heart rate, and subjective rating fatigue. However, heart rate alone is insufficient for monitoring fatigue. This is because heart rate is influenced by various physiological and behavioral factors such as cigarette smoking, mentally stressful situation, alcohol consumption, and energy drinks intake [47]. To address this issue, Aryal, Ghahramani and Becerik-Gerber [10] used heart rate in combination with human body thermoregulatory changes to monitor fatigue in construction workers. Hwang and Lee [13] showed the potential of using a wristwatch-based heart rate sensor to determine the levels of physical demands by measuring heart rate variability metrics. Maman, Yazdi, Cavuoto and Megahed [24] used jerk as a metric to derive from IMU sensors placed at the ankle, wrist, hip, and torso to detect physical fatigue. The study presents logistic regression models trained on the rating of perceived exertion. Jebelli, Choi and Lee [22] used PPG, EDA, and ST physiological signals of the worker in association with energy expenditure to determine the workers’ physical demands. The energy expenditure of the task was determined using the energy-expenditure prediction program (EEPP). However, several limitations exist in the current objective measurement systems for fatigue monitoring, such as the individual variability was not considered in the models, not applicable in a case where several tasks were performed in a short time, and most studies considered a subjective measurement of fatigue.

## 3. Proposed Fatigue Monitoring Framework

Figure 3 shows the proposed fatigue monitoring framework using forearm-based EMG and IMU data. The proposed system predicts the workers’ fatigue level by monitoring the aerobic fatigue threshold (AFT), which is the ratio of the average oxygen consumption to the activity-specific maximum aerobic capacity (MAC) value, as shown in Equation (1). The oxygen consumption and workers’ activity were predicted using forearm EMG and IMU data. Once the activity was recognized, the corresponding MAC value was obtained from the database to monitor AFT continuously. The proposed system can identify the activities and oxygen uptake every second. However, to predict the workers’ fatigue level, authors have considered the average AFT over five minutes. The predicted AFT using forearm EMG and IMU data was further validated using the data from metabolic analyzer and heart rate sensor. The proposed framework is highly suitable for the construction domain because it uses one single armband for data acquisition and uses the AFT metric, which is activity-dependent. Moreover, physiological signals such as EMG and IMU are activity-dependent and help in recognizing complex activities performed in a short interval of time.
(1)$$\mathrm{Aerobic}\;\mathrm{Fatigue}\;\mathrm{Threshold}=\frac{{\mathrm{Average}\;\mathrm{VO}}_{2}}{\mathrm{Activity}\;\mathrm{Specific}\;\mathrm{MAC}}$$

As shown in Figure 3, the proposed framework consists of six steps: (i) Data acquisition and preprocessing, (ii) activity recognition, (iii) oxygen uptake predictions, (iv) construction activity-specific MAC value, (v) continuous measurement of AFT, and (vi) workers’ fatigue monitoring. All these six steps were integrated to develop a real-time fatigue monitoring system. Each of these steps is discussed in detail in the following subsections.

### 3.1. Data Acquisition and Preprocessing

In the proposed framework, three wearable devices were used, namely Myo armband (EMG and IMU), metabolic analyzer, and heart rate monitor. The armband data was used to predict workers’ activities and AFT. However, the metabolic analyzer data was used as ground truth to develop the oxygen prediction model and heart rate for system validation. Since the data acquired from three devices were at different frequencies, data were preprocessed before feeding into the model. Later, the actual oxygen uptake and heart rate measured using a VO_2_ analyzer and Tickr heart rate monitor were at 1 Hz frequency, 289 statistical features were extracted from EMG and IMU raw data for every 1 s window. Since not all features add value to the model, only a few features were selected using feature selection techniques. For the activity classification, the top 100 features were selected using the SelectKBest with the ANOVA F-value function. However, for the oxygen uptake prediction, the features with Pearson’s correlation and mutual information greater than 0.1 were selected, there were 69 such features. Later, the selected EMG and IMU feature data was synchronized with VO_2_ (mL/kg/min), HR (bpm), and activity labels at 1 Hz frequency for ground truth. Finally, the data was normalized and standardized for oxygen uptake prediction and activity recognition models, respectively. After preprocessing, the ten participants’ data consisted of 48,515 samples. The input data was transformed into 3D shapes [100, 100, 14] and [48515, 100, 69] for activity recognition models and oxygen uptake prediction, respectively.

### 3.2. BiLSTM-Based Activity Recognition and Oxygen Uptake

This study proposes bidirectional long-short-term memory (BiLSTM)-based recurrent neural network for activity recognition and oxygen uptake prediction. The overall architecture of the proposed BiLSTM models is shown in Figure 4. The activity recognition and oxygen uptake prediction models consist of two BiLSTM layers, dropout layers and dense layers. Additionally, the Softmax activation function was used in the last layer of the activity recognition model. The categorical cross-entropy and MSE loss functions were used for training the classification and regression models. The models were trained using the ten participants’ data collected while performing fourteen scaffold-building activities. The leave-one-subject-out cross-validation (LOSO CV) was used to evaluate the performance of both models.

### 3.3. Construction Activity—Specific Maximum Aerobic Capacity (MAC)

In this study, authors have determined the construction activity-specific MAC value using a submaximal experiment protocol. The MAC value for four construction activities was determined by conducting submaximal experiments on ten participants. The MAC values of the four activities are shown in Table 1. For simplicity, the fourteen construction activities fall into one of these categories. For example, the carrying scaffold, crossbars, guardrail, and baseboard use the MAC value of carrying activity.

### 3.4. Aerobic Fatigue Threshold and Fatigue Monitoring

According to the National Institute of Safety and Health (NIOSH), the average oxygen during an eight hours workday is recommended to be no more than 33% VO_2max_. In other terms, the AFT value cannot exceed 33%. The forearm EMG and IMU data, activity recognition model, oxygen consumption model, and MAC values help continuously monitor AFT at a one-second level. Since the fatigue level rating was collected every five minutes, the average AFT was calculated every five minutes. Using the proposed system, the workers’ fatigue level can be assessed either by monitoring the AFT over the period or use of AFT variable to classify into one of the fatigue levels (i.e., None, Low, Moderate, High, and Very High—Fatigue).

## 4. System Feasibility Validation and Performance Evaluation

In order to test the feasibility of using AFT for assessing the fatigue level and for evaluating the performance of the automated fatigue monitoring system, collected oxygen uptake (VO_2_), heart rate (HR), and forearm inertial measurement unit (IMU) and electromyography (EMG) data from ten participants while performing simulated scaffold building activities.

### 4.1. Case Study of Scaffold Builder Activities

To evaluate the proposed fatigue monitoring framework, authors have considered one of the highly labor-intensive and repetitive construction activities, i.e., scaffold building. The scaffold-building activities involve complex body motions (free motion, repetitive motion, and impulsive motion) and different body parts (wrist, upper body, forearm, lower body, and whole body), which are commonly observed in various construction activities [48,49]. By observing the scaffold-building activities on construction sites, this study has considered fourteen scaffold-building activities, as shown in Table 2. The activities involve carrying, lifting, and installing various scaffold-related objects, such as scaffold frames, leveling jacks, guardrails, baseboards, and crossbars, which vary in weight and size. Other activities include walking, going up/down vertical ladders, hammering, and wrenching.

### 4.2. Experiment Setup

#### 4.2.1. Participants

Ten male active college students participated in this study (27 ± 1.70 years, 171.7 ± 4.13 cm, 76.70 ± 8.25 kg). All the participants were right-handed, non-smokers, and had no musculoskeletal disorders. Moreover, the activity level of the participants was moderate to vigorous. None of the participants had prior scaffold-building experience, but all the activities were demonstrated before the start of the experiment. After explaining the objective of the study and experiment procedures, written consent was obtained from the participants before starting the experiment. The experiment protocol consistent with the Declaration of Helsinki was reviewed and approved by the Institutional Review Board (IRB) at Louisiana State University (ID: IRBAM-20-0539).

#### 4.2.2. Measurements

To test the feasibility and performance of the proposed workers’ fatigue monitoring framework, three wearable sensors were used to collect the forearm IMU and EMG, oxygen uptake, and heart rate data while performing simulated scaffold-building activities. The forearm motion and muscle activity data were collected using the Myo armband developed by Thalmic Lab Inc., which captures IMU and EMG data at frequencies of 50 and 200 Hz [48]. The armband consists of 8-EMG electrodes and a nine-axis IMU sensor. A second-by-second oxygen uptake was measured using a portable metabolic analyzer, the VO_2_ Master Analyzer (VO_2_ Master Health Sensor Inc., Vernon, British Columbia, CA). Figure 5 shows the Myo armband and portable metabolic analyzer. In addition, the participant’s heart rate was recorded at a frequency of 1 Hz using a chest-strapped Wahoo Tickr heart rate monitor (Wahoo Fitness, Atlanta, GA, USA). Additionally, a rating of fatigue scale (ROF) on a scale of 0–10 was used to collect the participants’ perceived fatigue levels before and after the activity session [50]. Table 3 shows the rating of fatigue level and the corresponding label.

#### 4.2.3. Experiment Procedure

Once the written consent form was obtained from the participants, they were asked to warm up to prevent any injuries. Once the participant was ready, three sensors (metabolic analyzer, Myo armband, and heart rate monitor) were attached to the body. All the devices were calibrated for each participant, following the manufacturer’s guidelines. All the participants performed fourteen activities for five minutes each. The oxygen uptake, heart rate, and forearm IMU and EMG data were continuously recorded for each activity. The participants’ rating of fatigue (ROF) was collected before and after each activity (i.e., ROF value was captured every five minutes). The order of the activities was randomized for each participant. All the activities were performed in a warehouse environment at 72 F.

### 4.3. Data Analysis Protocol

#### 4.3.1. Aerobic Fatigue Threshold—Feasibility Validation and Performance Evaluation

Previous studies used aerobic fatigue threshold or exercise intensity to assess the worker capabilities or task workload evaluation [51,52,53]. Therefore, evaluating the use of AFT for workers’ fatigue monitoring is essential. Figure 6 presents the data analysis protocol to evaluate the feasibility and performance of AFT for fatigue monitoring. The best classifier was developed to predict the fatigue level on the unseen dataset using suitable features.

The oxygen uptake (VO_2_) and heart rate (HR) data were collected using a VO_2_ metabolic analyzer and Tickr chest strap HR monitor from ten participants (age: 27 ± 1.70 years, weight: 76.70 ± 8.25 kg, and height: 171.7 ± 4.13 cm) while performing scaffold building activities was used for this analysis. The oxygen uptake and heart rate data were recorded at 1 Hz frequency. Each participant performed fourteen scaffold-building activities listed in Table 2 for a maximum of five minutes or until they were exhausted. All the activities were randomized for each participant. In addition to VO_2_ and HR data, the participants’ rating of fatigue (ROF) was collected before and after each activity. The ROF was reported using the rating of the fatigue scale (0–10) and verbal anchors shown in Table 3 [10,50]. Using the rating of fatigue, the level of fatigue was assigned to one of the labels, i.e., none, low, moderate, high, and very high. To compare the performance of AFT to other fatigue assessment metrics such as heart rate (HR) [10] and percentage of HR reserve (%HRR) [13], the %HRR was calculated using Equation (2) for every one second.
(2)$$\mathrm{Percentage}\;\mathrm{of}\;\mathrm{HR}\;\mathrm{Reserve}\;(\%\mathrm{HRR})=\frac{{\mathrm{HR}}_{\mathrm{Working}}{-\;\mathrm{HR}}_{\mathrm{Resting}}}{{\mathrm{HR}}_{\mathrm{Maximum}}{-\;\mathrm{HR}}_{\mathrm{Resting}}}$$
where HR_Working_ = average working heart rate [bpm]; HR_Resting_ = resting heart rate [bpm]; and HR_Maximum_ = maximum heart rate is estimated using 220 age [bpm] [54,55].

The AFT, HR, and %HRR data obtained from ten participants for every one second were used to extract statistical features such as mean, minimum, maximum, and standard deviation for the duration of activity. In total, there are twelve features and 140 samples (10 participants × 14 activities). Since all the features are in different units, the features were normalized to scale all the features between zero and one. Later, the feature data was labeled with the level of fatigue (i.e., none, low, moderate, high, and very high) for each activity performed by the participant. There were no data samples with the “none” label.

The labeled feature data was further used to train ten commonly used machine learning-based classifiers, including Random Forest (RF), Decision Trees (DT), Naïve Bayes (NB), Linear Discriminant Analysis (LDA), Quadratic Discriminant Analysis (QDA), Support Vector Machine (SVM), Ada Booster (ADA), Logistic Regression (LR), K Nearest Neighbors (KNN), and Multilayer Perceptron (MLP). The classification analysis was performed using PyCaret—an open-source, low-code machine learning library in Python [56]. The models were evaluated using a 10-fold cross-validation technique, and the performance of the models was assessed using accuracy, precision, recall, and F1 Score. To evaluate the feasibility and performance of using AFT for fatigue level assessment, the classifier’s performance was compared for different feature combinations such as AFT, HR, %HRR, AFT + HR, AFT + %HRR, HR + %HRR, and AFT + HR + %HRR. However, the best fatigue level classifier was selected and used for further analysis on the unseen dataset.

#### 4.3.2. Fatigue Monitoring—Feasibility Validation and Performance Evaluation

Figure 7 presents the data analysis protocol to evaluate the feasibility and performance of the proposed fatigue monitoring framework. The authors have used the trained BiLSTM activity recognition model, MAC values for construction-specific activities, trained BiLSTM oxygen prediction model, and fatigue level classifier obtained using the ten participant data.

For the proposed system feasibility and performance evaluation, EMG and IMU data were collected from a participant (age = 29 years, height = 168 cm, weight = 75 kg, and Resting HR = 96) performing all fourteen scaffold building activities for approximately 85 min (5088 samples). The sequence of the activities and duration of the activities are shown in Table 4. Some of the IALJ, CDB, and IG activities are performed for a longer duration because they involve multiple tasks. For example, IALJ involves two tasks installing and adjusting leveling jacks performed for five minutes each continuously. In contrast, the activities such as GUDVL were performed for a short duration because the participant was completely exhausted after 1.40 min. In addition to armband data, VO_2_ and HR data were continuously recorded for the entire session. Moreover, a rating of the fatigue level was collected for every one minute.

First, the EMG and IMU data were preprocessed for activity recognition and oxygen uptake prediction. The oxygen uptake and heart rate data recorded every second were synchronized with the preprocessed EMG and IMU features. Additionally, the features dataset was labeled with actual activity ID for ground truth. Once the dataset was prepared, the trained activity recognition and oxygen uptake models developed in the previous chapters were implemented on unseen datasets to recognize activities and oxygen uptake for every one second. Using the model predictions and MAC values from Table 1, the authors determined AFT for every one second on the unseen dataset. The actual and predicted AFT values were analyzed using linear regression analysis to see how well the proposed system monitored AFT for one second, one-min, two-min, and over the entire activity duration.

For the fatigue level assessment, the predicted AFT values for every second were used to extract statistical features (i.e., mean, minimum, maximum, and standard deviation) for every one-min, two-min, and activity. The extracted AFT features were labeled with the subjective rating of fatigue level for ground truth. The best fatigue level classifier obtained in the previous section was used to predict the fatigue level of the unseen dataset. The actual and predicted fatigue levels were analyzed to see how well the predicted AFT can recognize workers’ fatigue levels compared to HR and %HRR.

## 5. Results

### 5.1. Aerobic Fatigue Threshold—Feasibility Validation and Performance Evaluation

First, the average AFT value for each fatigue level was estimated using the ten participants’ data, as shown in Table 5. The average AFT is above 33% for the high and very high fatigue levels where an individual is getting tired or very tired, which aligns with the NIOSH recommendation that an individual cannot sustain if AFT exceeds 33%. Moreover, Figure 8 shows the average AFT value for each activity where activities CPSF, CDB, and GUDVL are above the 33% threshold and align with the subjective fatigue rating rated as high or very high fatigue level activities. This shows that the subjective fatigue ratings are reliable for further analysis.

Further, the classification accuracy for the tested machine learning algorithms for different feature combinations is shown in Table 6. The results show that the highest classification accuracy was obtained using the decision tree classifier algorithm for AFT features. Also, the classification accuracy is highest for the features in combination with AFT (i.e., AFT + %HRR = 91.45%, AFT + HR + %HRR = 90.60, and AFT + HR = 90.60%). The highest classification accuracy for different feature combinations is highlighted in Table 6. Low classification accuracies were observed for HR (51.28%) and %HRR (50.43%) features. Similarly, the F1 Scores for the AFT (92.40%) are the highest compared to other features shown in Table 7. Moreover, Figure 9 shows the confusion matrix for the decision tree classifier using AFT features. It is observed that the model is classifying all four levels with 90% accuracy with a high misclassification rate of 11% between high and very high levels. From the accuracy, F1 Scores, and confusion matrix, the performance of the classifiers using AFT features is highest compared to HR and %HRR features. This analysis concludes that AFT features are highly suitable for assessing all four fatigue levels compared to %HRR and HR. The decision tree classifier built using the AFT features was further used to evaluate the fatigue levels of unseen data.

### 5.2. Fatigue Monitoring—Feasibility Validation and Performance Evaluation

The actual and predicted AFT on the unseen dataset for one second, one-min, two-min, and each activity over the entire experiment duration are shown in Figure 10, Figure 11 and Figure 12, respectively. The graphs show a peak trend for the high-intensity activities for all window sizes. The value of AFT varied based on the intensity and complexity of the activity. Figure 13 shows the average actual and predicted AFT for each activity, where the highest value was observed for GUDVL and the lowest for HAM activities. These results match the participants’ subjective fatigue rating. A linear correlation analysis was performed between actual and predicted AFT for one second, one-min, two-min, and each activity. The correlation results show that the highest coefficient of determination (R^2^) and root mean square error (RMSE) of 0.85 and 0.027 were observed for both one-min and two-min AFT. The lowest correlation was observed for one second, i.e., R^2^ = 0.71 and RMSE = 0.040. The goodness of fit curves for one second and five-min AFT is shown in Figure 14. The correlation analysis shows that the predicted AFT values have achieved a good fit and the variation in the AFT value for different activities demonstrates the feasibility of using predicted AFT values for fatigue assessment. Additionally, using a one-min or two-min average AFT helps to minimize errors and improves fatigue prediction accuracy.

Further, the predicted AFT features for one-min, two-min, and each activity are used for fatigue assessment to evaluate the performance of the proposed fatigue monitoring system. The classifier performance on predicted AFT features for one-min, two-min, and each activity is shown in Table 8. The results show that the overall performance of the fatigue assessment is better for the predicted AFT values for two-min (accuracy = 76.74%) compared to 1-min (accuracy = 71.05%) or for each activity (accuracy = 71.05%). However, the classification accuracy using HR (35.71%) and %HRR (35.71%) features of the unseen dataset is very low compared to AFT features. Figure 15 shows the high misclassification rate when using AFT features was observed between low and moderate fatigue levels.

Figure 16 and Figure 17 shows the actual and predicted fatigue level over the entire duration of the unseen dataset, where most of the time, the low and moderate fatigue levels were misclassified. Figure 18 shows that the fatigue level is accurately predicted for high-intensity activities such as GUDVL, CDB, and IBDL. Some low-intensity activities, such as ALJ, CLJ, and DG are misclassified as low instead of moderate. Moreover, it can be observed that the misclassifications of lower fatigue levels were less for two-min compared to one-min average AFT. This shows that the more extended window sizes are suitable for low-intensity activities. This indicates that the proposed system can be used for continuous monitoring of fatigue levels. 

Further, hypothesis testing was performed using a Chi-Squared test between actual and predicted fatigue levels for each window size (i.e., one-min, two-min, and average over each activity). The null hypothesis assumed that the actual fatigue levels are not related to predicted fatigue levels. The alternate hypothesis is that the actual fatigue levels are related to predicted fatigue levels. The results show that the *p*-value for all the window sizes is less than 0.05, therefore reject the null hypothesis and accept the alternate hypothesis, which concludes that there exists a significant relationship between actual and predicted fatigue levels for three window sizes (Table 9).

## 6. Discussion

This study confirmed that the proposed fatigue monitoring system could continuously assess workers’ fatigue levels. The use of the aerobic fatigue threshold (92.31%) to monitor the fatigue level has achieved high classification accuracy compared to the previous studies, which used energy expenditure (90%) [22] and skin temperature (80.60%) [10]. The better performance of the system is due to the high correlation of the AFT feature with the fatigue levels. Comparing the classification accuracy for different feature combinations shows that the AFT features have high performance compared to HR features, which agrees with the previous study [10]. This shows that the HR features alone are not suitable for fatigue level assessment for the activities performed in a short interval of time. Previous studies have considered two-min [10], 35 s [22], and 30 min [13] windows to assess the worker fatigue level. Jebelli et al. (2019) suggested that a higher window size is required to recognize the physical demand compared, the comparison of classification accuracy for different window sizes identified that two-min average AFT had achieved high classification accuracy compared to one-min over the duration of activity. This study recommends using a higher window size for low-intensity activities and a smaller window size for high-intensity activities. 

The AFT values for the high and very high fatigue level activities such as CDB, IBDL, and GUDVL are above 33%, which agrees with published guidelines for oxygen uptake [6,57]. Moreover, the classification accuracy was higher for high fatigue-level activities, which shows that AFT is a suitable metric to assess the fatigue level of high-intensity activities. The predicted AFT values using forearm motion and muscle activity data are highly correlated with actual values, showing that oxygen consumption is highly influenced by the type of activity performed in a short interval. The continuous monitoring of AFT can assess activity work severity classification based on published guidelines for oxygen uptake.

The proposed system is highly suitable for construction applications because it uses armband with 8-EMG electrodes and a nine-axis IMU to capture EMG and IMU data and AFT metrics, dependent on the activity. Since the EMG and IMU signals depend on activity, which helps recognize complex activities performed in a short time, the proposed system can be used for any trade. Moreover, the performance of the proposed system on the unseen dataset has shown the feasibility of using the system for complex high-intensity activities.

The previous studies classify the physical demand or fatigue level based on the physiological signal data from the worker. Unlike previous studies, the proposed system continuously measures the aerobic fatigue threshold using forearm EMG and IMU data that provides an opportunity to quantify the direct impacts of fatigue on accidents, evaluate the worker capabilities, and assess the workload evaluation of the task.

This study has successfully demonstrated forearm EMG and IMU data to monitor the workers’ fatigue level continuously; however, it has some limitations. The proposed system was validated and evaluated using only scaffold-building tasks. Even though the tasks were complex and highly physically demanding, different tasks should be studied for real-world fatigue monitoring applications. The study was conducted in a warehouse environment at 72 °F; however, other environmental and site conditions were not considered. Only one participants’ data was used to assess the performance of the proposed fatigue monitoring system. An extensive experiment needs to be performed with multiple participants of different ages, ethnicity, work experience, and physical health to evaluate the performance of the proposed system. The combined MAC value was used for hammering, wrenching, and dragging activities instead of individual activity-specific MAC value, which is one of the limitations of the study.

Future research focuses on improving the robustness and performance of the system by training the models using data from workers with different characteristics (i.e., age, work experience, ethnicity, and health conditions), trades, and working conditions. Moreover, investigate the deep learning algorithms to classify fatigue levels. AFT values range from extensive population data to training fatigue level classifiers to reduce the possible bias due to the subjective fatigue level rating.

## 7. Conclusions

This study proposes an automated framework to continuously monitor the worker fatigue level using forearm-based EMG and IMU sensors by measuring the aerobic fatigue threshold. The system validation and performance evaluation confirmed that the forearm EMG and IMU data could recognize complex construction activities, instantaneous oxygen uptake, continuous aerobic fatigue threshold, and classify fatigue level. The results conclude that AFT features could classify fatigue levels with a high accuracy of 92.31% compared to HR (51.28%) and %HRR (50.43%). Moreover, the results show that the AFT is greater than 33% for high and very high fatigue levels, which agrees with the NIOSH exercise intensity threshold. The proposed system is highly suitable for construction applications because the entire framework is dependent on the activity performed by the worker. Since the proposed framework is dependent on the activity, the system can be adaptable for any trade and site conditions. The continuous monitoring of fatigue levels helps assess the worker’s physiological status, evaluate the physical workload of the activity, quantify the direct impacts of the fatigue level on the accidents, and early detection of risk.

## Figures and Tables

**Figure 1 sensors-22-09729-f001:**
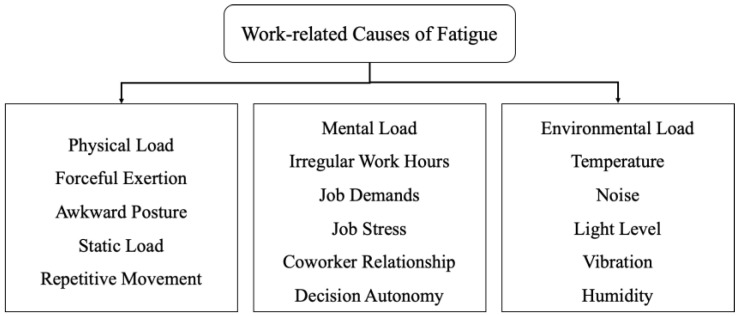
Work-related causes of physical fatigue.

**Figure 2 sensors-22-09729-f002:**
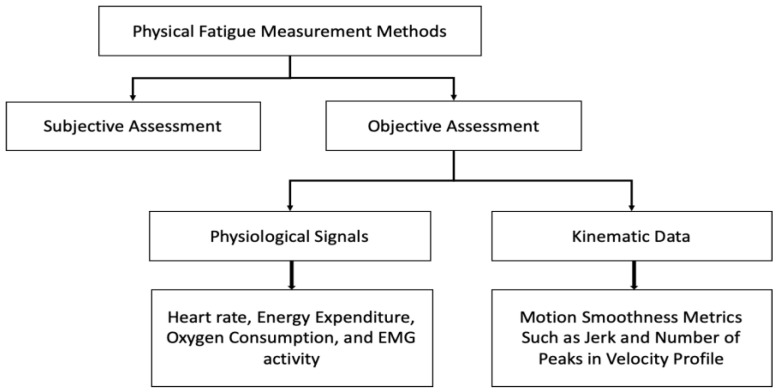
Physical fatigue measurement methods.

**Figure 3 sensors-22-09729-f003:**
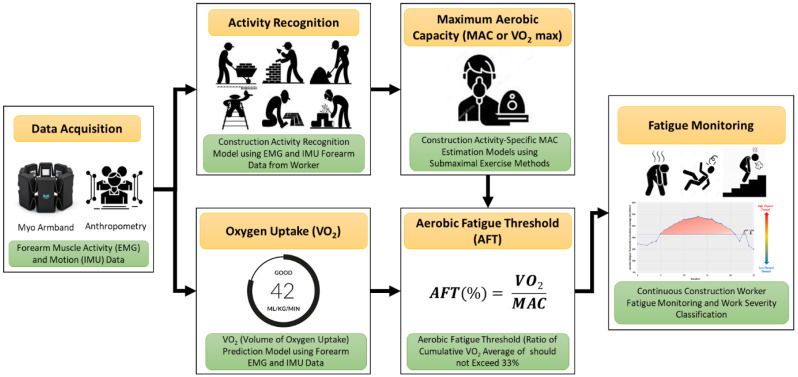
Proposed fatigue monitoring framework.

**Figure 4 sensors-22-09729-f004:**
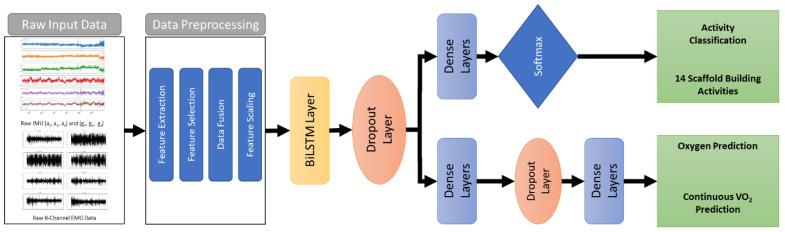
The overall architecture of the proposed BiLSTM model for activity recognition and oxygen prediction.

**Figure 5 sensors-22-09729-f005:**
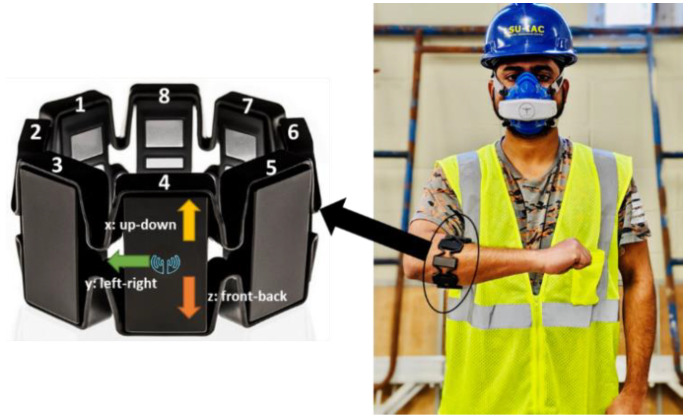
A participant wearing a forearm Myo armband and metabolic analyzer.

**Figure 6 sensors-22-09729-f006:**
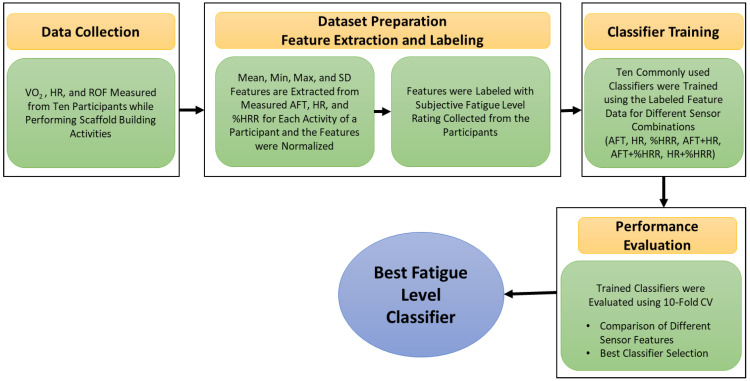
Data analysis protocol to evaluate the feasibility and performance of using the aerobic fatigue threshold for fatigue monitoring.

**Figure 7 sensors-22-09729-f007:**
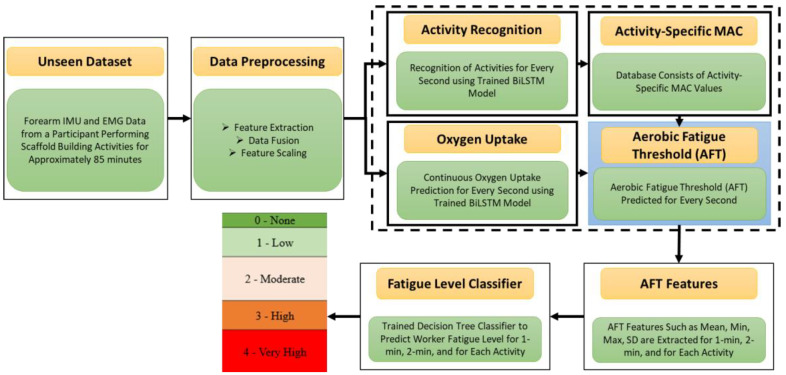
Data analysis protocol to evaluate the feasibility and performance of the proposed fatigue monitoring system.

**Figure 8 sensors-22-09729-f008:**
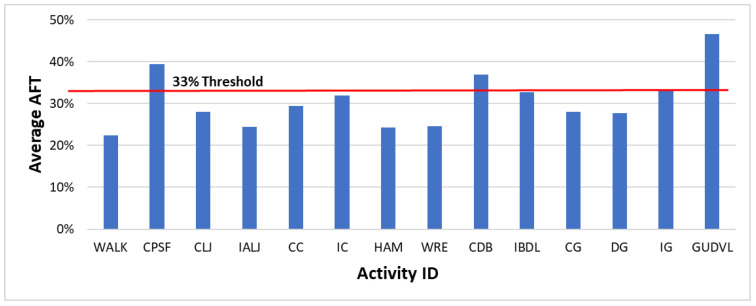
Average AFT for each activity.

**Figure 9 sensors-22-09729-f009:**
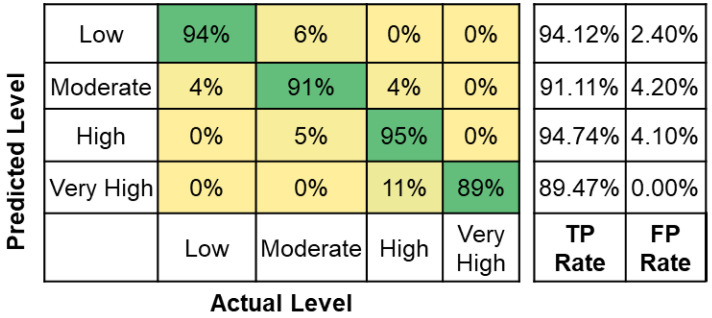
Confusion matrix for decision tree classifier using AFT features.

**Figure 10 sensors-22-09729-f010:**
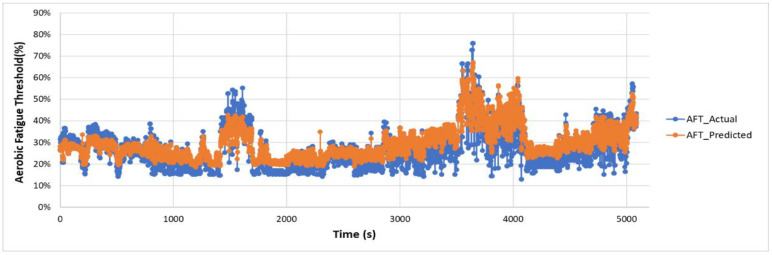
Actual and predicted AFT for every one second on the unseen dataset.

**Figure 11 sensors-22-09729-f011:**
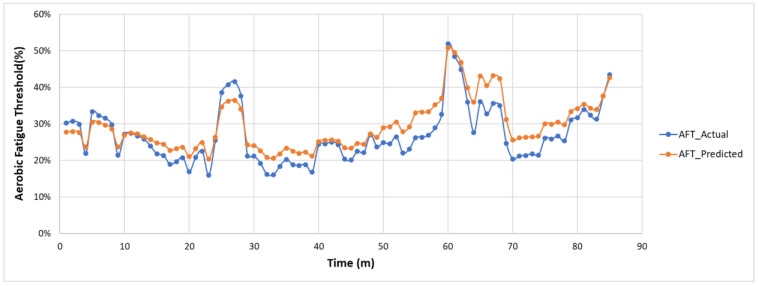
Actual and predicted AFT for every one-min on the unseen dataset.

**Figure 12 sensors-22-09729-f012:**
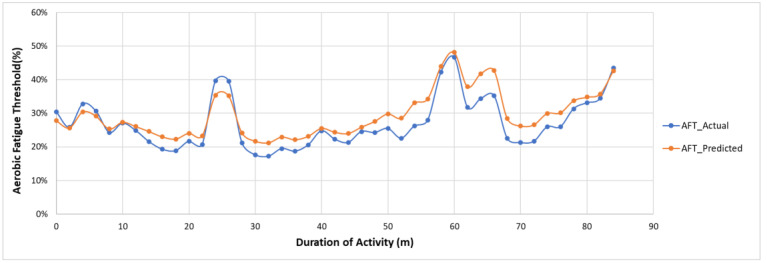
Actual and predicted AFT for every two-min on the unseen dataset.

**Figure 13 sensors-22-09729-f013:**
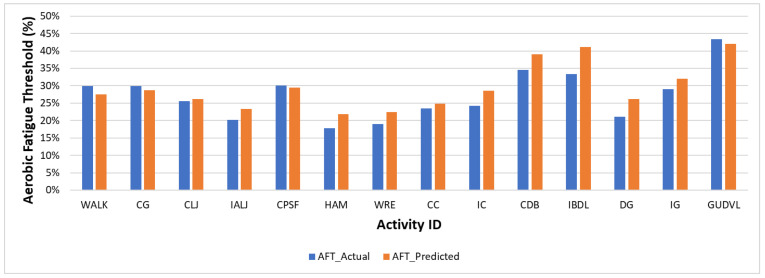
Actual and predicted AFT for each activity on the unseen dataset.

**Figure 14 sensors-22-09729-f014:**
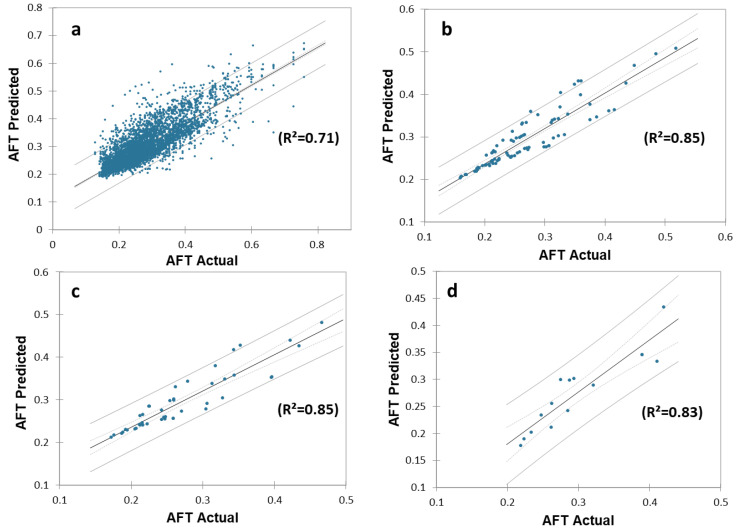
Correlation analysis for actual and predicted AFT values on the unseen dataset for (**a**) every one second, (**b**) average of one-min, (**c**) average of two-min, and (**d**) average for each activity.

**Figure 15 sensors-22-09729-f015:**
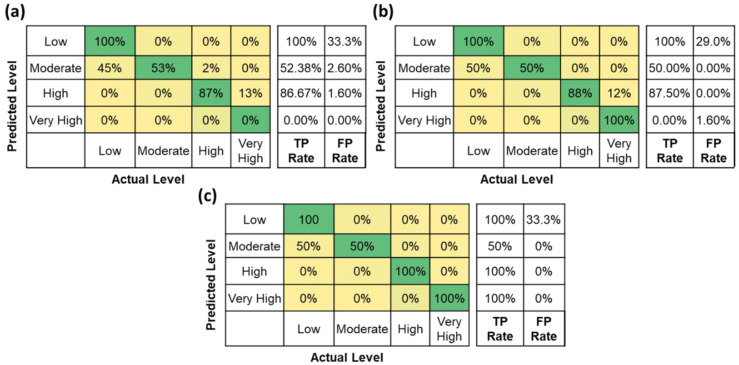
Confusion matrix of fatigue level assessment using predicted AFT features (**a**) for one-min, (**b**) for two-min, and (**c**) for each activity.

**Figure 16 sensors-22-09729-f016:**
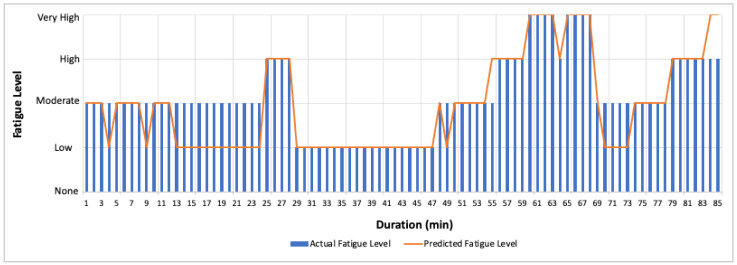
Predicted and actual fatigue level for every one-min on the unseen dataset.

**Figure 17 sensors-22-09729-f017:**
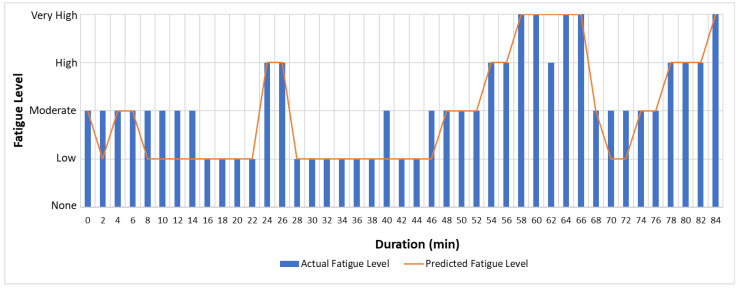
Predicted and actual fatigue level for every two-min on the unseen dataset.

**Figure 18 sensors-22-09729-f018:**
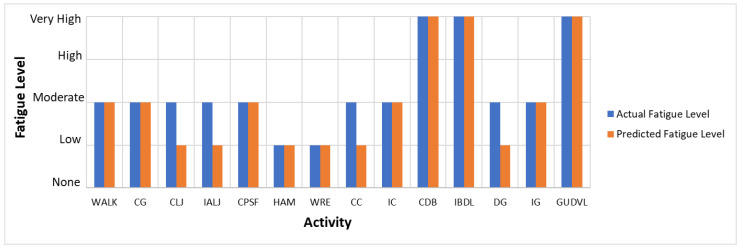
Predicted and actual fatigue level for each activity on the unseen dataset.

**Table 1 sensors-22-09729-t001:** MAC values for different construction activities.

Activities	MAC (L/min)	Scaffold Activities
Walking	2.946	WALK
Carrying	2.552	CPSF, CLJ, CC, CDB, CG
Lifting	2.816	IBDL
Combined	2.304	IALJ, IC, HAM, WRE, DG, IG, GUDVL

**Table 2 sensors-22-09729-t002:** Scaffold building activities.

SL. No.	Activities	Activity ID
1	Walking	WALK
2	Carrying/Positioning Scaffold Frame	CPSF
3	Carrying Leveling Jacks	CLJ
4	Inserting/Adjusting Leveling Jacks	IALJ
5	Carrying Crossbars	CC
6	Installing Crossbars	IC
7	Hammering	HAM
8	Wrenching	WRE
9	Carrying/Dragging Baseboard	CDB
10	Installing Baseboard on Different Levels	IBDL
11	Carrying Guardrail	CG
12	Dragging Guardrail	DG
13	Installing Guardrail	IG
14	Going Up/Down Vertical Ladder	GUDVL

**Table 3 sensors-22-09729-t003:** Fatigue rating scale and corresponding fatigue labels.

Fatigue Rating	Verbal Anchors	Fatigue Level & Labels
0	Not Fatigued at All	0—None
1	A Little Fatigued	1—Low
2		
3	Moderately Fatigue	2—Moderate
4		
5		
6	High Fatigue	3—High
7		
8	Very High Fatigue	4—Very High
9		
10		

**Table 4 sensors-22-09729-t004:** Activity sequence and duration of unseen dataset.

Sequence	Activity ID	Duration (min)
1	WALK	3.27
2	CG	4.98
3	CLJ	4.95
4	IALJ	10.03
5	CPSF	5.00
6	HAM	5.02
7	WRE	5.02
8	CC	5.00
9	CPSF	4.97
10	IC	5.02
11	CDB	10.10
12	IBDL	5.02
13	DG	5.02
14	IG	10.02
15	GUDVL	1.40

**Table 5 sensors-22-09729-t005:** Average AFT for each fatigue level.

Fatigue Level	Average AFT
Low	22.03%
Moderate	28.91%
High	36.00%
Very High	43.63%

**Table 6 sensors-22-09729-t006:** Classification accuracies for different feature combinations.

Model	AFT	%HRR	HR	AFT + %HRR	AFT + HR	HR + %HRR	AFT + HR + %HRR
RF	90.60%	44.44%	41.88%	89.74%	90.60%	44.44%	90.60%
DT	92.31%	41.03%	44.44%	91.45%	88.03%	34.19%	90.60%
NB	82.91%	46.15%	47.01%	79.49%	78.63%	42.74%	74.36%
LDA	87.18%	46.15%	51.28%	88.89%	83.76%	42.74%	82.91%
QDA	85.47%	38.46%	36.75%	79.49%	77.78%	37.61%	67.52%
SVM	72.65%	46.15%	44.44%	72.65%	66.67%	50.43%	68.38%
ADA	52.14%	46.15%	47.86%	52.14%	52.14%	46.15%	52.14%
LR	86.32%	50.43%	46.15%	79.49%	82.05%	45.30%	79.49%
KNN	81.20%	36.75%	38.46%	72.65%	72.65%	35.04%	64.96%
MLP	91.45%	46.15%	45.30%	88.89%	87.18%	46.15%	88.03%

**Table 7 sensors-22-09729-t007:** Classification F1 Score for different feature combinations.

Model	AFT	%HRR	HR	AFT + %HRR	AFT + HR	HR + %HRR	AFT + HR + %HRR
RF	90.60%	44.60%	42.10%	89.80%	90.60%	44.50%	90.70%
DT	92.40%	40.50%	43.20%	91.50%	88.10%	33.90%	90.60%
NB	82.80%	43.90%	42.20%	79.30%	78.20%	38.20%	73.90%
LDA	87.30%	44.80%	47.40%	88.90%	83.80%	41.60%	82.80%
QDA	85.30%	38.40%	36.30%	79.20%	77.30%	37.60%	65.50%
SVM	68.10%	46.20%	44.44%	70.70%	62.40%	50.40%	66.20%
ADA	52.10%	46.20%	47.90%	52.10%	52.10%	46.20%	52.10%
LR	86.30%	49.00%	42.60%	79.50%	82.10%	44.44%	79.60%
KNN	81.30%	36.20%	38.40%	72.90%	73.10%	35.30%	65.20%
MLP	91.50%	44.10%	43.50%	88.90%	87.20%	45.20%	88.20%

**Table 8 sensors-22-09729-t008:** Fatigue level assessment using predicted AFT features for every one-min, two-min, and each activity.

	Accuracy	Recall	Precision	F1 Score
Average Predicted AFT for 1-min	71.05%	71.10%	86.10%	72.40%
Average Predicted AFT for 2-min	76.74%	76.70%	86.10%	76.10%
Average Predicted AFT for Each Activity	71.43%	71.40%	90.50%	73.80%
HR Features for Each Activity	35.71%	35.70%	82.70%	36.20%
% HRR Features for Each Activity	35.71%	35.70%	40.50%	33.33%

**Table 9 sensors-22-09729-t009:** Chi-Squared test between the actual and predicted fatigue levels for different window sizes.

	Test Statistic	df	*p*-Value
1 min	160.93	9	2.20E-16
2 min	86.64	9	7.67E-15
Average Activity	30.333	9	0.000385

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
