# Peer review of "Automated and Continuous Fatigue Monitoring in Construction Workers Using Forearm EMG and IMU Wearable Sensors and Recurrent Neural Network"

_sensors, 2022, doi:10.3390/s22249729_

Round 1
Reviewer 1 Report
This research provides a comprehensive fatigue monitoring system using the wearable sensors. The topic is interesting and critical. In this paper, the gaps are well established, aims are clearly stated, methodology suits the objectives, and data collection and analysis have been carried out, and presented well. I believe this manuscript is of good quality and suitable for the journal. Some comments are listed.
(1) The Introduction is lengthy, and the authors use a lot of space to introduce the safety status and WMSDs, which are not the key point of this paper. Moreover, they put a quite detailed literature review here, while this repeats with the following section of Literature Review.
(2) Some sentences in Literature Review repeats with the Introduction. For instance, Lines 223-227 in Page5. In addition, the section of Pointe Departure is useless here and repeats with Introduction.
(3) The structure of Materials and Methods is confuse. I suggest the authors to introduce their fatigue motoring framework first and then give the experiment procedures.
(4) They use three wearable sensors, IMU and EMG, oxygen uptake, and heart rate data. They measure AFT based on IMU and EMG in the proposed monitoring framework, while the roles of oxygen uptake and heart rate are not clear in this paper. They put so many measures and experiments in this part, but the logic is confuse. I hope the author could re-organize the Section of Materials and Methods.
(5) Some errors: e.g., “eleven participants” in Page 289, I think, should be “ten participants”;
The title of 3.6.1 repeats with the title of 3.6.2.
Author Response
Please see the attached.
Thanks,
Srikanth Bangaru

Reviewer 2 Report
Dear Authors:
Congratulations to your article.
Thank you for the opportunity to review the manuscript "Automated and Continuous Fatigue Monitoring in Construction Workers using Forearm EMG and IMU Wearable Sensors and Recurrent Neural Network"
I was interested in reading this paper. This study aims to develop a system for continuous workers' fatigue monitoring by predicting aerobic fatigue threshold (AFT) using forearm muscle activity and motion data.
The topic being analysed seems to be relevant and I consider it original. The article does address a specific gap in the field. Since fatigue demonstrates itself in several complex processes, there is no single standard measurement method for fatigue detection.
The research adds to the subject area that the continuous monitoring of fatigue levels helps assess the worker's physiological status, evaluate the physical workload of the activity, quantify the direct impacts of the fatigue level on the accidents, and early detection of risk. The methodology is well thought out.
However, I have the following suggestions that you should review:
- You must renumber the citations in the text in consecutive order, and not jumping from one number to another.
- The sentences "Since the data acquired from the armband sensor (IMU and EMG), metabolic analyser (VO2), and heart rate monitor (HR) were at different frequencies (line 360), and "The data was preprocessed before being incorporated into the model (line 361), they must be unified into one.
- Line 512. Indicates Figure 7.5, when figure 7.5 does not exist. Are you referring to Figure 9?
- Line 529. Indicates Figure 7.9, when Figure 7.9 does not exist. Are you referring to Figure 13?
- Line 509. Indicates Table 7.5, when Table 7.5 does not exist. Please revise.
- Line 512. Indicates Table 7.6, when Table 7.6 does not exist. Please revise.
- Line 424. Indicates Table 7.2, when Table 7.2 does not exist. Please revise.
- You must write the references according to the standards of the journal.
- A scientific text should be written in the third person avoiding we.
The conclusions are consistent with the evidence and arguments presented and do they address the main question posed. The figures and tables are clear and reflect the information.
Sincerely
Author Response
Please see the attachment.
Thanks,
Srikanth Bangaru

Reviewer 3 Report
1. The introduction and literature review are both well-written and bring a well contextual situation and background knowledge. I do not have any point to complain about these subjects, congrats!
2. L.281 Why the purpose is in methods rather than the end of the introduction text?
3. L.333 The EMG uses a transducer rather than a sensor. Please fix the paragraph to assure that the reader knows about
4. L.360 Considering my last concern (point 3) if you agree with this, it is necessary to make an adjustment in all places that figure “armband sensor”. Feel free to choose a new way to nominate the armband.
5. L.419 and 425 Just fix subscript “VO2” à VO2
6. L.465, 470, and 472 Internationally we use “min” rather than “minute” when we have an Arabic numeral before the unit. 85 min, 1.4 min (why two decimal units?), and 1-min. I visualized that you used “min” in line 485. Let’s use like this in all paper, okay?
7. L. 583 Review the y-axis order in figure 16 – "none"; "low"; "high?"; "moderate?"; "very high".
Author Response
Please see the attachment.
Thanks,
Srikanth

Round 2
Reviewer 1 Report
All comments have been responded.